# Understanding patterns of fatigue in health and disease: protocol for an ecological momentary assessment study using digital technologies

Rosalind Adam  ,[1] Yojana Lotankar,[2] Corina Sas,[3] Daniel Powell,[4] Veronica Martinez,[5] Stephen Green,[5] Jonathan Cooper,[6] Katherine Bradbury,[7] Jonathan Sive,[8] Derek L Hill[9,10]

For numbered affiliations see end of article.

**Correspondence to**
Dr Rosalind Adam;
rosalindadam@abdn.ac.uk

## ABSTRACT

**Introduction** Fatigue is prevalent across a wide range of medical conditions and can be debilitating and distressing. It is likely that fatigue is experienced differently according to the underlying aetiology, but this is poorly understood. Digital health technologies present a promising approach to give new insights into fatigue.

The aim of this study is to use digital health technologies, real-time self-reports and qualitative interview data to investigate how fatigue is experienced over time in participants with myeloma, long COVID, heart failure and in controls without problematic fatigue. Objectives are to understand which sensed parameters add value to the characterisation of fatigue and to determine whether study processes are feasible, acceptable and scalable.

**Methods and analysis** An ecological momentary assessment study will be carried out over 2 or 4 weeks (participant defined). Individuals with fatigue relating to myeloma (n=10), heart failure (n=10), long COVID (n=10) and controls without problematic fatigue or a study condition (n=10) will be recruited. ECG patches will measure heart rate variability, respiratory rate, body temperature, activity and posture. A wearable bracelet accompanied by environment beacons will measure physical activity, sleep and room location within the home. Self-reports of mental and physical fatigue will be collected via smartphone app four times daily and on-demand. Validated fatigue and affect questionnaires will be completed at baseline and at 2 weeks. End-of-study interviews will investigate experiences of fatigue and study participation. A feedback session will be offered to participants to discuss their data.

Data will be analysed using multilevel modelling and machine learning. Interviews and feedback sessions will be analysed using content or thematic analyses.

**Ethics and dissemination** This study was approved by the East of England—Cambridge East Research Ethics Committee (22/EE/0261). The results will be disseminated in peer-reviewed journals and at international conferences.

**Trial registration number** NCT05622669.

## STRENGTHS AND LIMITATIONS OF THIS STUDY

⇒ This study integrates granular data from digital sensing technologies with participant self-reports and in-depth qualitative data.
⇒ Will use artificial intelligence techniques alongside multilevel modelling to detect patterns within temporal data.
⇒ Allows participants to reflect on their sensed data during bespoke feedback sessions, using qualitative data from these sessions to inform data interpretation.
⇒ Is not powered to quantify significant differences in fatigue ratings between groups of participants with different conditions.

Problematic fatigue is experienced by up to 98% of individuals with myeloma,[1] around 50% of those with heart failure[2] and over 70% of those with long COVID.[3] Fatigue is a multifaceted, fluctuating symptom which embodies biological, biochemical, physiological, psychological, emotional and behavioural processes.[4] Fatigue is distressing and causes reduced quality of life[5] and diminished economic productivity.[6]

Fatigue is considered as a single, poorly defined symptom, but fatigue as a symptom is likely to encapsulate multidimensional experiences and fatigue is unlikely to be a single entity. There is no one widely accepted definition of fatigue, but fatigue is often described as 'extreme and persistent mental and/or physical tiredness, weakness or exhaustion'.[7] Other definitions encompass the negative impact that fatigue can have on the individual, for example, an 'overwhelming feeling of sustained exhaustion that is debilitating and interferes with an individual's ability to function and perform activities'.[8]

There are likely to be specific patterns or characteristics of fatigue that vary according

## INTRODUCTION

Fatigue is a major clinical problem and is prevalent in most chronic medical conditions.

to the underlying mechanisms. For example, Powell *et al* showed that individuals with multiple sclerosis (MS) were more likely to have fatigue that peaked in the afternoon, that came on more suddenly and that was more likely to be present after physical activity compared with healthy controls.[9]

People with long COVID commonly describe triggers that worsen fatigue, such as physical activity, stress and sleep disturbance.[10 11] In people with heart failure, tissue hypoperfusion and a mismatch between catabolic and anabolic processes can cause skeletal myopathy.[12] This is thought to contribute to fatigue, particularly during physical exertion and fatigue is often experienced alongside breathlessness.[12] Fatigue is a common side effect of myeloma therapies, and during first treatment for myeloma, targeted medicines, chemotherapy agents and steroids are often given in cycles.[13] Fatigue is likely to vary during treatment cycles,[14] but this is underexplored. Specific patterns of fatigue in myeloma, heart failure and long COVID are under-researched.

Despite its high prevalence, fatigue is poorly understood.[15] Fatigue research can be challenging because fatigue is changeable, subjective and influenced by a wide range of individual, environmental and external factors. Fatigue also poses clinical challenges. When individuals present to clinicians with undifferentiated 'tiredness', standard diagnostic tests are usually unhelpful.[16 17] There are limited personalised treatment options for fatigue, and fatigue often remains untreated.[18]

Ecological momentary assessment (EMA)[19] is a method that is primarily focused on the investigation of within-person dynamic processes with high levels of ecological validity, using relatively intensive data collection techniques that monitor phenomena in real-time or close to real-time, while participants undertake their usual daily activities.[19] EMA methods have been used to gain insights into fatigue in several chronic medical conditions including renal failure,[20] MS[9] and cancer.[21] EMA studies have the advantage of minimising participant recall bias and highlighting temporal variations within data.[19] Few EMA studies have attempted to make comparisons between individuals with different clinical diagnoses.[22] Furthermore, advances in digital health technologies offer new opportunities to combine participant self-reports of symptoms with real-time objective physiological and environmental data.

This paper describes the protocol for an EMA study of fatigue. Digital health technologies will be used to capture in-depth objective physiological, activity, self-report and environmental measurements from individuals with myeloma, long COVID, heart failure and a control group without these conditions. Cancers, infectious diseases and cardiovascular diseases account for a significant global disease burden.[23] The exemplar conditions were chosen because they involve fatigue as a core or highly prevalent symptom, and because they are likely to involve different underlying mechanisms of fatigue and different patterns of fatigue. They are also conditions in which we expect to see fluctuations in the fatigue experience over time that might be observable, predictable and explainable (eg, according to medication cycle in myeloma, relating to postexertional malaise in long COVID or activity levels in heart failure).

The aim of the study is to measure fatigue within individuals over time, to quantify variations in fatigue levels within individuals and to describe relationships between patient-reported fatigue, patient-reported triggers for fatigue and sensed parameters including activity levels, sleep, heart rate variability (HRV) and other physiological parameters. Physiological parameters such as HRV could give important insights into the fatigue experience. For example, HRV is the fluctuation in time intervals between each heartbeat[24] and is linked to autonomic nervous system activation.[25] HRV can give insights into emotions and stress.[25] It has been suggested that HRV could be linked to fatigue severity in chronic fatigue syndrome.[26] HRV has also been studied as a marker of driver fatigue.[27] In this study, qualitative and quantitative data will be combined to investigate differences in the fatigue experience within and between individuals over time.

This study is exploratory and will test the feasibility, acceptability and practicalities of deploying the digital technologies and of the study procedures with a view to conducting a large-scale investigation into replicable, measurable, quantitative differences in fatigue patterns between groups of individuals with different medical conditions. The study will also develop artificial intelligence (AI) algorithms and explore the value of AI techniques in analysing and interpreting EMA study data.

## METHODS
### Study design and setting
This EMA study will be conducted remotely, enrolling participants from anywhere within the UK. Participants will self-rate their physical and cognitive fatigue four times daily and on demand over two or 4 weeks (according to participant preference). The study is an exploratory feasibility study, and the primary objective is to use both quantitative and qualitative methods to characterise lived experiences of fatigue and to explore temporal patterns in fatigue within individuals with different medical conditions.

### Outcomes
The primary outcome is descriptive and concerns how patient reports of fatigue change over time, and the temporal relationships between fatigue, self-reported triggers for fatigue and sensed parameters (see below).

Secondary outcomes relate to the feasibility of the study procedures. Secondary outcomes include ability to recruit eligible participants at the required rate; attrition rate; suitability of the study questionnaires and patient reported outcome measures; acceptability and desirability of the study procedures; usability of the technologies; logistics of remote study assessments and data

collection; and potential predictors of missing self-report data during the EMA study.

## Sample size

The study will aim to enrol 40 participants, including 10 participants with myeloma, 10 with heart failure, 10 with long COVID and 10 control participants. It may be the case that the sample sizes in the groups are not evenly matched. This will be determined by participant interest and response. It is also possible that some participants provide limited data or have high levels of missing data (eg, due to sensor failure or poor adherence to EMA ratings). The research team may decide to recruit additional participants into a group in which participants have high levels of missing data, where missing data precludes meaningful insights into the participant's fatigue experience, and this will be judged during the study by the lead researcher.

This is a feasibility study without a quantitative primary outcome measure. As such, a formal sample size calculation is not required; however, the sampling design has been chosen to sufficiently capture the expected within-day variability in fatigue following discussions with patient partners. In parallel group pilot trials, a sample size of 12 participants per group is suggested as a 'rule of thumb' that will give adequate data to inform future definitive studies.[28] The EMA study design employed here differs from traditional parallel group studies in that multiple observations are taken in the same individual over time, allowing for detailed within person descriptions of the phenomenon and temporal relationships/associations. In this study, granular patient reported outcome data will be combined with rich qualitative data. The estimate of ten patients per group is a pragmatic one that considers the primary study objective, researcher capacity, study duration, number of daily EMA ratings and availability of sensors and other technical equipment.

## Eligibility criteria

Inclusion and exclusion criteria are listed in table 1. Eligible participants are adults with fatigue (considered by them to be worse than 'normal tiredness') related to myeloma, long COVID or heart failure and control group participants without a study condition who do not experience problematic fatigue (worse than 'normal tiredness'). Participants will not be given a specific definition of fatigue at the start of the study, and no cut-off points will be used for the severity of fatigue or its impact on function. Instead, the experience of fatigue will be documented in detail for every participant using the combination of validated, sensed and self-report measures described below.

## Recruitment and consent

Eligible patients with myeloma will be invited to participate by clinicians from the haematology clinic at University College London Hospital. Eligible patients with heart failure will be invited to participate by clinicians working in National Health Service (NHS) Grampian's heart failure nursing service. Participants with heart failure will also be recruited from General Practices in NHS Grampian, facilitated by the NHS Research Scotland Primary Care network.

Participants with myeloma, heart failure or long COVID who took part in previous focus groups as part of this project and who have consented to being contacted about future related research will be invited to participate by email. Relevant charity and patient advocacy organisations will also be invited to share study information leaflets with their members.

Control group participants will be recruited via staff mailing lists at collaborating academic institutions, patient involvement groups and from community walking organisations.

The study may be advertised on X (formerly known as Twitter) if recruitment targets are not achieved via other routes.

Individuals identified by NHS clinicians will be handed an invitation letter, information sheet, reply slip and reply-paid envelope by their clinicians and asked to contact the study team if they are interested in participating. Those recruited by general practices will be identified by searches of electronic medical records and sent invitation packs by post. Those invited by email will receive the participant information sheet and invitation letter as an attachment and will be asked to reply by email to the study team to indicate their interest.

Interested individuals will have an initial telephone or Teams video call, during which a researcher will explain the study and answer any questions. Eligibility screening will be conducted, and for eligible participants who wish to participate, audio-recorded verbal consent will be taken. A paper consent form will be initialled and signed by the researcher, and a copy will be sent to the participant. A copy of the participant consent form is provided in online supplemental file 1.

Participants will be offered a £50 voucher as a token of thanks.

## Participant timeline and data collection methods

The participant timeline is summarised in figure 1.

### Baseline assessments

At study baseline, participants will complete the Mental and Physical State and Trait Energy and Fatigue Scales (STEF) parts one and three[29–31]; Positive and Negative Affect Scales for 'past few weeks' time frame[32] and the Modified Fatigue Impact Scale.[33] These questionnaires were chosen as they have been validated and administered together, they will give insights into both physical and mental fatigue, 'usual' fatigue (as a complement to the EMA measures), emotions and the impact of fatigue on the individual.

### Sensors

Participants will be posted a study kit containing an android smartphone with data SIM, an ECG patch

**Table 1** Inclusion and exclusion criteria

| Group allocation | Inclusion criteria | Exclusion criteria |
|---|---|---|
| All participants | Willing to participate in intermittent ecological momentary assessments of fatigue, to wear an ECG patch, to provide questionnaire responses and to participate in an end-of-study interview. | Difficulty communicating in English. Adults lacking capacity to consent. Under 18 years of age. Declines to participate. Under investigation for or starting treatment for an endocrine, metabolic or thyroid condition where the participant has not been established on a stable therapeutic dose of a licensed therapy for that condition. A confirmed diagnosis of sleep apnoea or narcolepsy. Hospital Anxiety and Depression Scale (HADs) score at baseline greater than 8 on the depression questions,[36] which might indicate untreated or undertreated depression. Shift work that involves overnight working between 21:00 and 9:00. |
| Group A, individuals with myeloma | A confirmed diagnosis of myeloma. Has experienced fatigue that is perceived by the participant to be worse than 'normal tiredness' and that they associate with myeloma or treatment for myeloma. | Uncontrolled hypercalcaemia. Current or previous diagnosis of heart failure or long COVID. An active primary cancer diagnosis other than myeloma. |
| Group B, individuals with heart failure | A formal diagnosis of heart failure. All stages of heart failure and all aetiologies and with no specific ejection fraction cut-off. Has experienced fatigue that is perceived by the participant to be worse than 'normal tiredness' and that they associate with their cardiac disease or its treatment. | A current or previous diagnosis of myeloma or long COVID. Active cancer. |
| Group C, individuals with long COVID | Experiencing fatigue that is perceived by the participant to be worse than 'normal tiredness', with or without other physical or psychological symptoms that developed during or after an infection consistent with COVID-19. The fatigue (plus or minus any other symptoms) has continued for ≥12 weeks and is not explained by an alternative diagnosis. | A current or previous diagnosis of myeloma or heart failure. Active cancer. |
| Group D, control group | Individuals aged 18 years or above without the disease conditions specified in groups A–C. | Presence of myeloma or another active cancer, heart failure or long COVID. One or more chronic medical conditions which are unstable, poorly controlled and perceived by the individual to be causing fatigue. Persistent or severe fatigue symptoms that are perceived by the individual to be worse than 'normal tiredness'. Taking sedating medications to manage anxiety or insomnia including but not limited to benzodiazepines or 'Z' drugs, zopiclone, zolpidem and others in this British National Formulary Class. |

(Vital Patch, Vital Connect, California, USA), a wearable bracelet containing accelerometer accompanied by four environment beacons (Panoramic Digital Health SAS, Grenoble France).

A wide range of technical options were considered to capture the data required in this study. A selection of these and the factors that were considered when choosing the technologies are presented in online supplemental file 2. The technologies were discussed within the whole study team and consensus was reached about the technologies to deploy.

The VitalPatch ECG is an approved medical device and will be worn for the first 7 days of the study. The patch measures respiratory rate, a single lead ECG with RR interval, heart rate parameters (including HRV), body temperature, activity levels (actimetry) and body position.

The ECG patch communicates by Bluetooth with a dedicated app (MediBioSense, UK) preinstalled on the study phone.

The Panoramic digital health bracelet (Panoramic Digital Health SAS, Grenoble, France) is a wrist worn device containing a three-axis accelerometer, three-axis magnetometer, three gyroscopes, temperature sensor and atmospheric pressure sensor to measure physical activity and sleep. The bracelet is used alongside four Bluetooth beacons, which are placed within labelled positions within the participants' homes (eg, kitchen, living room, bedroom, top of stairs). The beacons communicate with the bracelet and generate data about the wearer's proximity to each beacon, and also record temperature, sound level and light level in their location, providing environmental context to the bracelet sensor data.

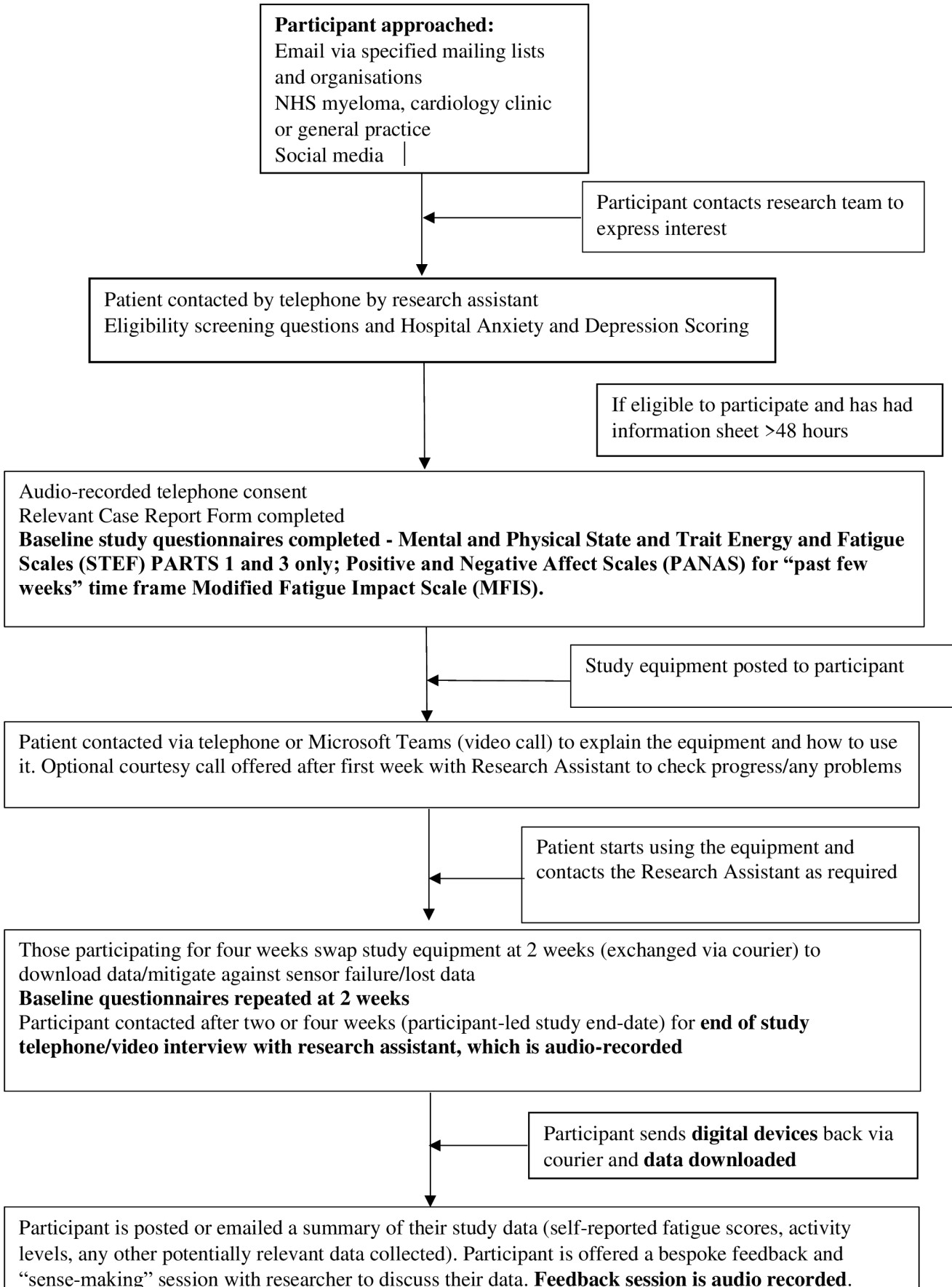

**Participant approached:**
Email via specified mailing lists and organisations
NHS myeloma, cardiology clinic or general practice
Social media

Participant contacts research team to express interest

Patient contacted by telephone by research assistant
Eligibility screening questions and Hospital Anxiety and Depression Scoring

If eligible to participate and has had information sheet >48 hours

Audio-recorded telephone consent
Relevant Case Report Form completed
**Baseline study questionnaires completed - Mental and Physical State and Trait Energy and Fatigue Scales (STEF) PARTS 1 and 3 only; Positive and Negative Affect Scales (PANAS) for "past few weeks" time frame Modified Fatigue Impact Scale (MFIS).**

Study equipment posted to participant

Patient contacted via telephone or Microsoft Teams (video call) to explain the equipment and how to use it. Optional courtesy call offered after first week with Research Assistant to check progress/any problems

Patient starts using the equipment and contacts the Research Assistant as required

Those participating for four weeks swap study equipment at 2 weeks (exchanged via courier) to download data/mitigate against sensor failure/lost data
**Baseline questionnaires repeated at 2 weeks**
Participant contacted after two or four weeks (participant-led study end-date) for **end of study telephone/video interview with research assistant, which is audio-recorded**

Participant sends **digital devices** back via courier and **data downloaded**

Participant is posted or emailed a summary of their study data (self-reported fatigue scores, activity levels, any other potentially relevant data collected). Participant is offered a bespoke feedback and "sense-making" session with researcher to discuss their data. **Feedback session is audio recorded**.

**Figure 1** Participant timeline and flow chart. (NHS = National Health Service).

**Box 1    Self-rating prompts within the m-Path app**

1. Overall, how fatigued do you feel at the moment?
• I feel no fatigue, strongest feeling of fatigue ever (0–10 slider).
2. How do you feel right now with regard to your capacity to perform your typical activities.
PHYSICAL ACTIVITIES…
• I feel no fatigue—0–10 slider—strongest feelings of fatigue ever felt.
3. How do you feel right now with regard to your capacity to perform your typical activities.
MENTAL ACTIVITIES…
• I feel no fatigue—0–10 slider —strongest feelings of fatigue ever felt.
4. Is your fatigue better, the same or worse than usual?
CONDITIONAL
• If the same—no further questions.
• If better—question about factors making better.
• If worse—question about factors making worse.
Factors making better:
Do you think that any of the following might have affected your current levels of fatigue (select all that apply)?
• Exercise or physical activity within the last day.
• Exercise or physical activity more than a day ago.
• Taking a medicine (please specify).
• Taking rest.
• Taking caffeine.
• Less emotions or stress.
• My mood.
• Spending time with other people.
• The weather.
• Quiet time.
• Something else (please specify)—free text option.
Factors making worse:
Do you think that any of the following might have affected your current levels of fatigue (select all that apply)?
• Exercise or physical activity within the last day.
• Exercise or physical activity more than a day ago.
• Taking a medicine (please specify).
• Emotions or stress.
• My mood.
• Tasks or work relating to my employment.
• Tasks or work relating to my home life.
• Spending time with other people.
• The weather.
• Noise.
• Other (please specify).

## EMA assessments

Participants will use the m-path app (https://m-path.io), which is a General Data Protection Regulation (GDPR)-compliant research tool developed by the Katholieke Universiteit Leuven. The app will be preloaded onto the study smartphone, or participants can opt to download this on their own device if they prefer. The app will be used to send participants short questionnaires four times daily to rate their cognitive fatigue and physical fatigue from 0 to 10, and to note any associated triggers for improvements in or deteriorations in fatigue. The anchoring and wording of the self-rating questions is based on 'state' fatigue items from the Mental and Physical STEF Scale.[29]

A full summary of the self-rating questions is provided in text Box 1.

M-path will send audible alerts to the smartphone at predetermined intervals, spaced throughout the day and preagreed with the participant to fit with work or personal commitments. Participants will be able to miss self-reports and will be asked to add extra reports on-demand if they notice that their fatigue levels are better or worse than usual.

## End-of-study assessments

Participants complete all baseline questionnaires again after 2 weeks and participate in an end-of-study qualitative interview, conducted according to a schedule (online supplemental file 3). They are invited to participate in an optional feedback and 'sense-making' session, during which a researcher provides them with feedback on the data provided and any interesting features of the data are discussed. Feedback sessions are audio recorded and used in analyses.

Any adverse events relating to the use of technology or participating in the study will be recorded. Any serious adverse events will be reported to the sponsor.

## Data management and analysis

ECG patch data will be downloaded manually from the phone by the research team when the phone is returned. Raw encrypted data from the Panoramic bracelets are also downloaded directly from the bracelets by the research team, where data are decrypted, quality checked and analysed. Periods of non-wear are detected using the temperature sensor and accelerometer. The raw accelerometer data are processed using the open-source OxWearables algorithms to measure step count per minute,[34] sleep and level of activity.[35] The beacon data are processed to label the processed bracelet sensor data with location in the home, by identifying the beacon with the highest Bluetooth received signal strength indicator. Visual inspection of the output of these algorithms alongside the Vital-Patch output can be viewed in a data annotator to identify features of interest in the data including timing of getting out of bed, timing of any stair climb and timing of any period of immobility following movement in the home. m-Path app data are downloaded from an on-line study dashboard

Summary data can be generated from the Panoramic digital health platform, including: total sleep time, step count per hour over the day, time spent and activity in each room per day, time spent and activity when away from home. More granular data can also be determined including time taken to get out of bed, number of room transitions per day, average daily time climbing stairs (where applicable), cadence for walking and total period of immobility following movements within the home, and whether the participant gets out of bed when they wake up at night (eg, to go to toilet or kitchen) or stay in bed. Data on directional change (eg, turning/changing

walking direction), change in height (eg, going upstairs, walking up slopes) can also be generated

Patient feedback reports will be created manually, identifying potentially interesting or unexplained features from self-report and sensed data, for example, particularly high or low self-reported fatigue scores, or rapid changes in fatigue scores. Visualisations will be produced (format and content will be iterative and refined during the study based on participant feedback). Visualisations will show fatigue self-rating scores superimposed on charts showing date, time, activity levels, sleep patterns and location within the home or being out of the home. The reports will also provide short narrative summaries about the participants' fatigue self-ratings and patterns that are being seen within the data. Feedback sessions with DLH (professor of digital health), CS (professor of human–computing interaction) or RA (academic general practitioner) will be audio recorded and used to give insights into the face validity of the sensed data and the fatigue experience more broadly. Participants will not be given medical or self-management advice.

AI algorithms will be created by VM and SG (University of Cambridge) to automatically handle multivariety/multidimensional data, detect symptom fluctuation over time and identify patterns, trends and groups. New algorithms using machine learning (ML) will be developed and tested to understand patterns of fatigue and relationships between sensed parameters. Within-person and between-person and within-group and between-group differences will be analysed using multi-level modelling and ML. Despite the limited number of participants to be enrolled in this study, the high frequency of both the sensor data and EMA responses makes it possible to build exploratory AI models that link features derived from the sensor data and the EMA responses. Models will be exploratory but could identify potential relationships between the sensor data and EMA responses that could inform the design of future studies and larger scale AI models

Qualitative data from interview transcripts will be analysed using Framework and thematic analysis.

Multiple investigators will be involved in the analyses. Pseudonymised data will be shared using a data sharing platform approved by the University of Aberdeen. The final dataset will be available to investigators whose proposed use of the data has been approved by the research ethics committee and who are parties to the project data sharing agreement.

## Patient and public involvement

Six on-line focus groups were conducted with individuals with fatigue (qualitative results will be reported separately). Participants with experience of a study condition gave insights into their lived experience of fatigue and their thoughts about using digital technologies to measure aspects of the fatigue experience. The results from the focus groups directly informed study design, including study duration. Some participants would find

4 weeks of data collection onerous, whereas for others, 2 weeks of data collection would be insufficient to capture perceived patterns in their fatigue (eg, during myeloma treatment cycles).

The focus groups also led to the inclusion of the participant feedback session. It was important to participants that their sensed and self-reported data were explained to them and that they were given the opportunity to reflect on this data with an experienced researcher.

A patient partner with myeloma liaised with the study team during the preparation of the study documents and gave feedback on the protocol and patient information sheet.

## Potential implications

This is an exploratory feasibility study but an ambition for the work is that we start to identify discrete patterns of fatigue in different individuals. Identifying different fatigue signatures could lead to more effective classification of fatigue. Ultimately, a fatigue classification system could help with diagnosis of unexplained or undifferentiated fatigue and to tailor different management approaches.

The feasibility study will also provide practical details about the usability, reliability, acceptability and utility of the sensors. Feedback sessions with the participants will help to gauge face validity of the measurements, and which, if any, sensed parameters are meaningful to people with fatigue. Objective digital measurements that correlate well with subjective experiences in people with fatigue could enhance clinical trials of drug and non-drug treatments for fatigue both by providing objective inclusion criteria and endpoints for efficacy. The measurements might also have implications for fatigue self-management. Digital measurements could be used to objectively identify factors (eg, activity patterns, sleep patterns, diurnal variations) that improve or worsen fatigue levels and that could potentially be modified. The measurements may be used in the future to help to inform pacing, goal setting and other self-management approaches to fatigue management.

## Ethics and dissemination

The study is sponsored by the University of Aberdeen. It was approved by the East of England—Cambridge East Research Ethics Committee (22/EE/0261). The study team are interdisciplinary researchers spanning primary care (clinical), health psychology, engineering, medical physics, human computing interaction and AI. The results will be disseminated in a range of peer-reviewed journals spanning these disciplines and at international conferences.

## Study status

Participant recruitment commenced on 7 December 2022 and was completed in December 2023. 40 participants were consented. Feedback sessions with the researchers are ongoing and new data are still being generated from

these sessions. Data analysis is in the early stages. Data collection is expected to be complete by September 2024. The approved protocol is V.2.0, 8 December 2022.

**Author affiliations**
[1]Academic Primary Care, Institute of Applied Health Sciences, University of Aberdeen, Aberdeen, UK
[2]Psychology and Neuroscience, University of Westminster, London, UK
[3]School of Computing and Communications, Lancaster University, Lancaster, UK
[4]Health Psychology, Institute of Applied Health Sciences, University of Aberdeen, Aberdeen, UK
[5]Department of Engineering, University of Cambridge, Cambridge, UK
[6]James Watt School of Engineering, University of Glasgow, Glasgow, UK
[7]Psychology, University of Southampton, Southampton, UK
[8]University College London Hospitals NHS Foundation Trust, London, UK
[9]Medical Physics & Biomedical engineering, University College London Faculty of Engineering Sciences, London, UK
[10]Panoramic Digital Health Ltd, Grenoble, France

**Acknowledgements** The research team would like to thank all patient participants in our on-line focus groups for their input into the design of this study and to the National Health Service clinicians who are assisting with participant identification and study invitations.

**Contributors** RA: contributed to conceptualisation, funding acquisition, methodology, writing—original draft preparation and writing—review and editing. YL contributed to project administration and writing—review and editing. CS contributed to conceptualisation, funding acquisition and writing—review and editing. DP contributed to methodology and writing—review and editing. VM contributed to conceptualisation, funding acquisition and writing—review and editing. SG contributed to writing—review and editing. JC contributed to conceptualisation, funding acquisition and writing—review and editing. KB contributed to conceptualisation, funding acquisition and writing—review and editing. JS contributed to writing—review and editing. DLH contributed to conceptualisation, funding acquisition and writing—review and editing.

**Funding** This work is being funded by Engineering and Physical Sciences Research Council (EPSRC). Grant reference EP/W003228/1. RA is funded by a Chief Scientist Office (Scotland) Senior Clinical Academic Fellowship (reference SCAF/18/02). Neither the funders nor the sponsor had any input into the design of this study and will have no role in data analysis.

**Competing interests** DLH, professor of Digital Health at UCL, is also founder and CEO of Panoramic Digital Health, a company that is providing technology for this study. This potential conflict of interest is documented in accordance with UCL Disclosure of Conflict and Declaration of Interest Policy and has been reviewed by the Sponsor. None of the other collaborators have any conflicts to declare.

**Patient and public involvement** Patients and/or the public were involved in the design, or conduct, or reporting, or dissemination plans of this research. Refer to the Methods section for further details.

**Patient consent for publication** Not applicable.

**Provenance and peer review** Not commissioned; externally peer reviewed.

**ORCID iD**
Rosalind Adam http://orcid.org/0000-0003-3082-6578

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
