## [Reviewer comments · BMJ Open]

ARTICLE DETAILS

TITLE (PROVISIONAL)	Understanding patterns of fatigue in health and disease: Protocol for an Ecological Momentary Assessment Study using digital technologies
AUTHORS	Adam, Rosalind; Lotankar, Yojana; Sas, Corina; Powell, Daniel; Martinez, Veronica; Green, Stephen; Cooper, Jonathan; Bradbury, Katherine; Sive, Jonathan; Hill, Derek

VERSION 1 – REVIEW

REVIEWER	Jun, Jeehye Chung-Ang University
REVIEW RETURNED	11-Dec-2023

GENERAL COMMENTS	Thank you for providing me with this excellent protocol manuscript. This is a very good research design to understand patterns of fatigue using digital technologies in four different groups. In the introduction section, the authors systematically and clearly addressed the research gap in the area, which made me be interested in this topic. The methodology and statistical approach are well-identified. Just one suggestion, I think that Figure 1 is too narrative to understand the study timeline. Please condense the contents and clearly indicate the study timeframe and what to access (i.e., measures) at each time point. I have no further comments or suggestions for this study protocol. I look forward to the findings of this study. Thank you.
---

REVIEWER	Finsterer, Josef Krankenanstalt Rudolfstiftung, Vienna
REVIEW RETURNED	05-Jan-2024

GENERAL COMMENTS	Understanding patterns of fatigue in health and disease: Protocol for an Ecological Momentary Assessment Study using digital technologies Before including patient it must be clarified what the cause of fatigue is. Only homogenous groups should be examined.
---

REVIEWER	El-Toukhy, Sherine National Institutes of Health
REVIEW RETURNED	15-Mar-2024

GENERAL COMMENTS	Review of "Understanding patterns of fatigue in health and disease: Protocol for an ecological momentary assessment study using digital
---

	technologies.” Below are some comments that I hope the authors find useful as they revise the protocol.  - Can you elaborate on the implications of this work? With more frequent measures of fatigue, do you anticipate a change in medication regimens or recommendations that we give patients? - Given that the nature and level of fatigue can differ by disease, what is your definition of fatigue and its threshold in this study? - Please justify the relationship between fatigue and heart rate variability. - What is the nature of feedback that will be given to participants in the sense-making session? Is it medical in nature? And if so please clarify who from the research team will be providing the feedback. Are there certain elements that will be provided for each participant?
--	---

REVIEWER	Mills, Sarah University of St Andrews School of Medicine
REVIEW RETURNED	18-Mar-2024

GENERAL COMMENTS	Fatigue is an important, common and under-researched medical topic. The paper discusses fatigue in patients with myeloma, heart failure or long COVID and the authors have mentioned that this is due to them having different mechanisms of fatigue, which is true but which would be true of other medical conditions as well. It would be beneficial for the authors to include a more detailed explanation of why these symptoms were included. In lines 36-52 the authors have given a brief overview of the impact of fatigue on each of the chosen conditions. There are still only five papers cited in this literature review section. It would be ideal to expand this section further to give more specifics of what is already known about the patterns of fatigue in these different conditions. If there are no further relevant publications, due to this being a very under-researched field, the authors could include a statement to that effect. The methodology is sound and appropriate, and will provide a reasonable sample size for this feasibility study. EMA is an excellent choice for investigating fatigue and the authors display a strong understanding of its application. Outcomes are appropriately chosen and well explained. Sample size is ambitious for a feasibility study but the authors are experienced and will likely meet recruitment targets. In the absence of a sample size calculation (which is appropriate given the lack of a quantitative primary outcome measure), it would be ideal to know whether the authors intend to continue to recruit until thematic saturation has been reached, or if they would consider alternative choices for if/when to expand their number of participants.
--

	It would be ideal to have further detail and references discussing the EMA study design and what typical recruitment numbers for EMA research are (p7 lines 38-47). Eligibility criteria are clear and well-presented. Recruitment and consent is detailed and includes a mix of traditional and modern methods. Participants are being reimbursed at rates at or above NIHR INVOLVE rates, which is commendable. A discussion of how the applicants will ensure recruitment from people with protected or marginalised characteristics, or how they will interpret the qualitative data if their participant group is not reflective of the wider population, would be ideal to include. The timeline is clear and well-presented. The data collection methods are thoroughly set out and explained. Page 10 line 49 please remove the extra full stop It would be ideal in page 10-11 to add in references for the tools that are discussed eg Panorama digital health bracelet, to allow the reader to familiarise themselves with the tools used. Data management plan is well written and complete. More detail of the AI methods that will be developed in this project would be useful here. PPI strategy is well explained and sufficient to ensure representation and maximise input. Study Status - states recruitment will be completed by 18 December 2023, which has now passed. Please could this section be updated? Funding statement - please spell out EPSRC before abbreviating Competing interests Given DH's conflict of interest it would be good to discuss what decision-making underpinned the selection of the technology used in this study, and what alternative technologies were considered and rejected, in order to demonstrate maximum transparency. References References 13 and 22 appear incomplete, please review.
--	---

VERSION 1 – AUTHOR RESPONSE

REVIEWER ONE

Comment: Thank you for providing me with this excellent protocol manuscript. This is a very good research design to understand patterns of fatigue using digital technologies in four different groups. In the introduction section, the authors systematically and clearly addressed the research gap in the area, which made me be interested in this topic. The methodology and statistical approach are well-identified.

Response: Thank you.

Comment: Just one suggestion, I think that Figure 1 is too narrative to understand the study timeline. Please condense the contents and clearly indicate the study timeframe and what to access (i.e., measures) at each time point.

Response: Thank you for the suggestion. We have now condensed the figure, reducing the narrative text in the Figure, and highlighting the data collection time points in Bold font. We trust that this will make the study timeline easier to understand.

REVIEWER TWO

Comment: Before including patient it must be clarified what the cause of fatigue is.

Response: Definitely determining what causes fatigue in any one individual is limited by the complexity of the human fatigue experience, the multifaceted and multidimensional nature of fatigue, and the fact that diagnostic tests for fatigue are unhelpful in around 95% of people in primary care. Nevertheless, we know that the conditions we chose to study are characterized by high levels of fatigue. We are confident that we are correctly identifying individuals with the relevant study conditions by involving NHS clinicians in recruiting most of our heart failure and myeloma participants, and by following a rigorous screening process at study baseline. There is no single accurate diagnostic test for long COVID and we rely on participant self-identification of typical symptoms developing after an infection consistent with COVID. This is the case with most studies of long COVID, and we are using long COVID as a model of post-viral fatigue.

We only include individuals who are confident that their fatigue is related to their relevant clinical diagnosis i.e. myeloma, heart failure, or long COVID. Fatigue is subjective and the participant is often best placed to judge whether their diagnosed condition causes them to feel fatigued.

In this exploratory study we collect detailed qualitative data about fatigue experiences in addition to physiological sensed data and longitudinal data. Analysis is at an early stage, but we are beginning to see interesting patterns in the groups. For example, people with myeloma experience predictable sleep changes during their chemotherapy cycle, followed by changes in sensed and self-reported fatigue parameters. By studying fatigue in granular detail for up to four weeks and integrating detailed qualitative reports, we can be confident that we are capturing as much about the “cause” of fatigue (e.g. medicines, sleep disturbance, relationship with activity) as we can across these different and important study conditions, and this is a strength of the study design.

Comment: Only homogenous groups should be examined

Response: Every effort has been made to keep our study groups as homogenous as possible with respect to the diagnosed conditions. We excluded those with concomitant depression or metabolic/endocrine conditions that were not adequately treated, which could have caused heterogeneity and potential confounding. In any exploratory study pragmatic judgements must be made to ensure that we are capturing a breadth of different experiences, and that we are able to recruit enough participants to give us meaningful insights into fatigue. This feasibility study will help us to judge whether inclusion/exclusion criteria are optimal for any future large-scale trials.

REVIEWER THREE

Comment: Can you elaborate on the implications of this work? With more frequent measures of fatigue, do you anticipate a change in medication regimens or recommendations that we give patients?

Response: We have added a “Potential Implications” section at the end of the manuscript as follows:

“Potential Implications

This is an exploratory feasibility study but an ambition for the work is that we start to identify discrete patterns of fatigue in different individuals. Identifying different fatigue signatures could

lead to more effective classification of fatigue. Ultimately, a fatigue classification system could help with diagnosis of unexplained or undifferentiated fatigue and to tailor different management approaches.

The feasibility study will also provide practical details about the usability, reliability, acceptability and utility of the sensors. Feedback sessions with the participants will help to gauge face validity of the measurements, and which, if any, sensed parameters are meaningful to people with fatigue. Objective digital measurements that correlate well with subjective experiences in people with fatigue could enhance clinical trials of drug and non-drug treatments for fatigue both by providing objective inclusion criteria, and endpoints for efficacy. The measurements might also have implications for fatigue self-management. Digital measurements could be used to objectively identify factors (e.g. activity patterns, sleep patterns, diurnal variations) that improve or worsen fatigue levels, and that could potentially be modified. The measurements may be used in the future to help to inform pacing, goal setting, and other self-management approaches to fatigue management.”

Comment: Given that the nature and level of fatigue can differ by disease, what is your definition of fatigue and its threshold in this study?

Response: The reviewer makes an important point. Fatigue is a poorly defined symptom, and the definitions and understanding of fatigue can differ between individuals and between researchers from different disciplines. We have added extra detail about the definitions of fatigue to the introduction section of the manuscript as follows:

“Fatigue is considered as a single, poorly defined symptom, but fatigue as a symptom is likely to encapsulate multidimensional experiences and fatigue is unlikely to be a single entity. There is no one widely accepted definition of fatigue, but fatigue is often described as “extreme and persistent mental and/or physical tiredness, weakness or exhaustion” (7). Other definitions encompass the negative impact that fatigue can have on the individual, for example, an “overwhelming feeling of sustained exhaustion that is debilitating and interferes with an individual’s ability to function and perform activities” (8)”

Given the lack of consensus in the literature about a definition for fatigue, the subjective nature of the experience, and the fact that this study seeks to explore fatigue in granular detail over up to four weeks, we allowed participants to judge whether they were suffering from fatigue at baseline. We did not give the participants a formal definition except that they should be suffering from fatigue that “is worse than normal tiredness” and relating to the condition that they were suffering (e.g. long COVID, myeloma or heart failure). During the study, we were able to formalise our understanding of the participants’ experiences of fatigue and it may be that work from this project helps us to better define fatigue and the heterogeneity and variability of fatigue in different individuals. The following text has been added to the methods section of the manuscript:

“Participants will not be given a specific definition of fatigue at the start of the study and no cut-off points will be used for the severity of fatigue or its impact on function. Instead, the experience of fatigue will be documented in detail for every participant using the combination of validated, sensed and self-report measures described below.”

In addition, in our suggested change to the manuscript in response to reviewer three’s first comment, we made clear that this work could have potential implications on defining meaningful thresholds of fatigue.

Comment: Please justify the relationship between fatigue and heart rate variability.

Response: Additional detail has been added to the introduction section as follows:

“The aim of the study is to measure fatigue within individuals over time, to quantify variations in fatigue levels within individuals, and to describe relationships between patient-reported

fatigue, patient-reported triggers for fatigue, and sensed parameters (including activity levels, sleep, heart rate variability, and other physiological parameters). Physiological parameters such as heart rate variability could give important insights into the fatigue experience. For example, Heart Rate Variability (HRV) is the fluctuation in time intervals between each heartbeat (22) and is linked to autonomic nervous system activation (23). Heart rate variability can give insights into emotions and stress (23). It has been suggested that HRV could be linked to fatigue severity in chronic fatigue syndrome (24). HRV has also been studied as a marker of driver fatigue (25). In this study-qualitative and quantitative data will be combined to investigate differences in the fatigue experience within and between individuals over time.”

Comment: What is the nature of feedback that will be given to participants in the sense-making session? Is it medical in nature? And if so please clarify who from the research team will be providing the feedback. Are there certain elements that will be provided for each participant?

Response: No medical advice is offered in the feedback sessions. The idea of including feedback sessions in this study came from patient public engagement work. People with fatigue told the study team that they would like to “see” their data to understand what we had been able to measure, but also to get insights into their patterns of fatigue, exercise, sleep, etc. The feedback sessions are proving to be extremely valuable to the research team (and popular with the participants). We are able to highlight interesting features in the data and get the participants’ explanation of what we are seeing. For example, a participant with myeloma was noted to be sleeping out-with the home environment on a certain week-day afternoon. He was able to confirm that he was indeed sleeping and that this was a hospital chemotherapy day. We are able to meaningfully explain features in our data and to gauge face validity. The following text has been added to the manuscript:

Patient feedback reports will be created manually, identifying potentially interesting or unexplained features from self-report and sensed data, for example particularly high or low self-reported fatigue scores, or rapid changes in fatigue scores. Visualisations will be produced (format and content will be iterative and refined during the study based on participant feedback). Visualisations will show fatigue self-rating scores superimposed on charts showing date, time, activity levels, sleep patterns, anlocation within the home or being out of the home. The reports will also provide short narrative summaries about the participants’ fatigue self-ratings and patterns that are being seen within the data. Feedback sessions with DH (Professor of Digital Health), CS (Professor of Human-Computing Interaction), or RA (academic General Practitioner) will be audio-recorded and used to give insights into the face validity of the sensed data and the fatigue experience more broadly. Participants will not be given medical or self-management advice.

REVIEWER FOUR

Comment: Fatigue is an important, common and under-researched medical topic. The paper discusses fatigue in patients with myeloma, heart failure or long COVID and the authors have mentioned that this is due to them having different mechanisms of fatigue, which is true but which would be true of other medical conditions as well. It would be beneficial for the authors to include a more detailed explanation of why these symptoms were included.

Response: The reviewer makes a good point. Fatigue is such a prominent symptom in so many conditions that we could have chosen from a wide variety of chronic conditions. At the outset we did consider conditions like inflammatory arthritis, multiple sclerosis and head injury. As principle investigator I was keen to include examples of a cardiovascular disease, a cancer, and a post-infectious presentation because together these types of conditions cause the greatest burden to societies throughout the world. We decided to include conditions that could lead to fatigue through discrete mechanisms, and conditions in which we could identify and recruit relatively homogenous groups. Another consideration was that we wanted to study conditions where fatigue would be likely

to fluctuate during the study period, so that we could observe changes during study participation and identify potential triggers and patterns. We have updated the introduction text as follows:

This paper describes the protocol for an EMA study of fatigue. Digital health technologies will be used to capture in-depth objective physiological, activity, self-report, and environmental measurements from individuals with myeloma, long COVID, heart failure, and a control group without these conditions. Cancers, infectious diseases, and cardiovascular diseases account for a significant global disease burden (22). These exemplar medical conditions have been chosen because they involve fatigue as a core or highly prevalent symptom, and because they are likely to involve different underlying mechanisms of fatigue, and different patterns of fatigue. They are also conditions in which we expect to see fluctuations in the fatigue experience over time that might be observable, predictable and explainable (for example, according to medication cycle in myeloma, relating to post-exertional malaise in long COVID or activity levels in heart failure).

Comment: In lines 36-52 the authors have given a brief overview of the impact of fatigue on each of the chosen conditions. There are still only five papers cited in this literature review section. It would be ideal to expand this section further to give more specifics of what is already known about the patterns of fatigue in these different conditions. If there are no further relevant publications, due to this being a very under-researched field, the authors could include a statement to that effect.

Response: The following statement has been added to the introduction:

“Specific patterns of fatigue in myeloma, heart failure and long COVID are under-researched.”

Comment: The methodology is sound and appropriate, and will provide a reasonable sample size for this feasibility study. EMA is an excellent choice for investigating fatigue and the authors display a strong understanding of its application.

Response: Thank you.

Comment: Outcomes are appropriately chosen and well explained.

Response: Thank you.

Comment: Sample size is ambitious for a feasibility study but the authors are experienced and will likely meet recruitment targets.

Response: We are confident that we will meet recruitment and data collection targets.

Comment: In the absence of a sample size calculation (which is appropriate given the lack of a quantitative primary outcome measure), it would be ideal to know whether the authors intend to continue to recruit until thematic saturation has been reached, or if they would consider alternative choices for if/when to expand their number of participants.

Response: This study generates a huge quantity of data for each individual (heart rate data, activity, sleep, temperature, patient reported outcome, self-ratings and qualitative data). Thematic saturation was not a consideration in this study, which is exploratory and integrates both quantitative and qualitative analyses. It is also important to highlight that qualitative data is generated from both the study interviews and the feedback sessions with participants, giving rise to two detailed qualitative accounts of fatigue per participant. The feedback sessions are fairly novel and generate unique insights into the fatigue experience. We are confident that a sample size of 40 will answer our research questions.

Comment: It would be ideal to have further detail and references discussing the EMA study design and what typical recruitment numbers for EMA research are (p7 lines 38-47).

Response: We have added the following extra detail with references:

Ecological momentary assessment (EMA) (19) is a method that is primarily focussed on the investigation of within-person dynamic processes with high levels of ecological validity, using relatively intensive data collection techniques that monitor phenomena in real-time, or close to real-time, whilst participants undertake their usual daily activities (19).

We have also added a reference for a comprehensive systematic review of EMA (Perski O et al, Understanding health behaviours in context: a systematic review and meta-analysis of ecological momentary assessment studies of five key health behaviours. Health Psychology Review 2022) to the introduction.

In terms of sample size for EMA studies, there is a degree of complexity. Sample size will vary based on the outcome of interest, the number of self-ratings over time, how stable/unstable the phenomenon is, the duration of the study and other important factors. It is beyond the scope of our introduction to be able to explain the complexity of sample size determination in EMA. In the scoping review of the use of EMA in cancer (referenced in our introduction - Kampshoff C et al. Ecological momentary assessments among patients with cancer: a scoping review Eur J Cancer Care 2019), 12 EMA studies were identified in cancer and the number of participants in these studies ranged from 15 to 107 (mean 53). In the systematic review of EMA by Perski et al, the mean number of participants across over 600 studies was 100. Again, there is a large variation in sample size between studies and it is difficult to give a simple summary of typical sample size.

Comment: Eligibility criteria are clear and well-presented.

Response: Thank you.

Comment: Recruitment and consent is detailed and includes a mix of traditional and modern methods. Participants are being reimbursed at rates at or above NIHR INVOLVE rates, which is commendable. A discussion of how the applicants will ensure recruitment from people with protected or marginalised characteristics, or how they will interpret the qualitative data if their participant group is not reflective of the wider population, would be ideal to include.

Response: In this feasibility study we have no specific mechanisms to ensure recruitment of people with protected or marginalised populations, but as a feasibility study we will fully report participant demographics and representativeness. This will help us to target recruitment to ensure diversity and representation of any groups of participants that we may be missing/that are under-represented in the next stages of the project.

Comment: The timeline is clear and well-presented. The data collection methods are thoroughly set out and explained.

Response: Thank you.

Comment: Page 10 line 49 please remove the extra full stop

Response: Thank you for noticing the error. We have removed the extra full stop.

Comment: It would be ideal in page 10-11 to add in references for the tools that are discussed e.g. Panorama digital health bracelet, to allow the reader to familiarise themselves with the tools used.

Response: We currently have two papers on the sensing technologies that are in preparation (one is under peer review) but not yet published. We will be able to reference these in the main study report.

Comment: Data management plan is well written and complete. More detail of the AI methods that will be developed in this project would be useful here.

Response: The AI methodology in this study is exploratory and a separate manuscript is being prepared to detail the potential contribution of AI techniques to sensed data and EMA data and to outline potential models. We have updated the methods section in this protocol as follows:

“Despite the limited number of participants to be enrolled in this study, the high frequency of both the sensor data and EMA responses makes it possible to build exploratory AI models that link features derived from the sensor data and the EMA responses. Models will be exploratory but could identify potential relationships between the sensor data and EMA responses that could inform the design of future studies and larger scale AI models.”

Comment: PPI strategy is well explained and sufficient to ensure representation and maximise input.

Response: Thank you

Comment: Study Status - states recruitment will be completed by 18 December 2023, which has now passed. Please could this section be updated?

Response: We have updated this section as follows:

Participant recruitment commenced on 7th December 2022 and was completed in December 2023. Forty participants were consented. Feedback sessions with the researchers are ongoing and new data are still being generated from these sessions. Data analysis is in the early stages. Data collection is expected to be complete by the September 2024.

Comment: Funding statement - please spell out EPSRC before abbreviating

Response: We have completed this change.

Comment: Given DH's conflict of interest it would be good to discuss what decision-making underpinned the selection of the technology used in this study, and what alternative technologies were considered and rejected, in order to demonstrate maximum transparency

Response: This is important and the approach to choosing the most appropriate technological solutions was rigorous. We have uploaded a supplementary data file that lists some of the technologies considered and the factors that were considered by the team when choosing the technologies. We have also added the following text to the methods section:

A wide range of technical options were considered to capture the data required in this study. A selection of these and the factors that were considered when choosing the technologies are presented in Supplementary data file 2. The technologies were discussed within the whole study team and consensus was reached about the technologies to deploy.

Comment: References 13 and 22 appear incomplete, please review.

Response: Thank you for noticing and we have updated the references.